# Geography shapes the phylogeny of frailejones (Espeletiinae Cuatrec., Asteraceae): a remarkable example of recent rapid radiation in sky islands

Mauricio Diazgranados[1] and Janet C. Barber[2,3]

[1] Natural Capital and Plant Health Department, Wakehurst Place, Royal Botanic Gardens, Kew, Ardingly, West Sussex, United Kingdom
[2] Department of Biology, Saint Louis University, Saint Louis, MO, United States
[3] Missouri Botanical Garden, Saint Louis, MO, United States

Corresponding author
Mauricio Diazgranados,
M.Diazgranados@kew.org

## ABSTRACT

**Background**. The páramo ecosystem, located above the timberline in the tropical Andes, has been the setting for some of the most dramatic plant radiations, and it is one of the world's fastest evolving and most diverse high-altitude ecosystems. Today 144+ species of frailejones (subtribe Espeletiinae Cuatrec., Asteraceae) dominate the páramo. Frailejones have intrigued naturalists and botanists, not just for their appealing beauty and impressive morphological diversity, but also for their remarkable adaptations to the extremely harsh environmental conditions of the páramo. Previous attempts to reconstruct the evolutionary history of this group failed to resolve relationships among genera and species, and there is no agreement regarding the classification of the group. Thus, our goal was to reconstruct the phylogeny of the frailejones and to test the influence of the geography on it as a first step to understanding the patterns of radiation of these plants.

**Methods**. Field expeditions in 70 páramos of Colombia and Venezuela resulted in 555 collected samples from 110 species. Additional material was obtained from herbarium specimens. Sequence data included nrDNA (ITS and ETS) and cpDNA (rpl16), for an aligned total of 2,954 bp. Fragment analysis was performed with AFLP data using 28 primer combinations and yielding 1,665 fragments. Phylogenies based on sequence data were reconstructed under maximum parsimony, maximum likelihood and Bayesian inference. The AFLP dataset employed minimum evolution analyses. A Monte Carlo permutation test was used to infer the influence of the geography on the phylogeny.

**Results**. Phylogenies reconstructed suggest that most genera are paraphyletic, but the phylogenetic signal may be misled by hybridization and incomplete lineage sorting. A tree with all the available molecular data shows two large clades: one of primarily Venezuelan species that includes a few neighboring Colombian species; and a second clade of only Colombian species. Results from the Monte Carlo permutation test suggests a very strong influence of the geography on the phylogenetic relationships. Venezuelan páramos tend to hold taxa that are more distantly-related to each other than Colombian páramos, where taxa are more closely-related to each other.

**Conclusions**. Our data suggest the presence of two independent radiations: one in Venezuela and the other in Colombia. In addition, the current generic classification will need to be deeply revised. Analyses show a strong geographic structure in the phylogeny, with large clades grouped in hotspots of diversity at a regional scale, and

in páramo localities at a local scale. Differences in the degrees of relatedness between sympatric species of Venezuelan and Colombian páramos may be explained because of the younger age of the latter páramos, and the lesser time for speciation of Espeletiinae in them.

## INTRODUCTION

The mechanisms of evolution and drivers of diversity in tropical ecosystems are not well understood, and most existing studies have focused on lowland taxa and ecosystems. The Andes are one of the most topographically and climatically complex orographic systems (*Killeen et al., 2007*; *Särkinen et al., 2012*) and they are a renowned hotspot for biodiversity (*Brooks et al., 2006*; *Young et al., 2015*). Above the timberline in the tropical Andes, the páramos dominate the landscape. They are considered the world's most diverse high-altitude ecosystem (*Luteyn, 1999*; *Rangel-Ch, 2000*; *Sklenár et al., 2005*), and one of the fastest evolving ecosystems (*Madriñán, Cortés & Richardson, 2013*). With an estimated age of 2–4 million-years (*Hooghiemstra & Van der Hammen, 2004*), the páramos emerge on top of the Andean cordilleras and massifs, rising in some instances to the snow line above 4,700 m. In a few cases, they appear in small humid mountain refugia with poorly drained soils surrounded by high Andean forest. Whether they are isolated by deep Andean valleys or by dense forest, biogeographically páramos function like islands and are often referred to as "sky islands."

*Cuatrecasas (1934)* classified the páramos in three zones based on the type of vegetation and the altitudinal gradient: (1) subpáramos or low páramo, have thickets and shrubby vegetation (~3,000–3,600 m); (2) páramo proper, are open grasslands of heliophilous plants (~3,600–4,200 m); and (3) superpáramo, the highest plant-life zone of these neotropical mountains, with soil affected by frequent frost and scarce psychrophilic vegetation (~4,200–4,800 m). The páramos have been the stage for great diversification of several groups of organisms, including amphibians, mosses, and vascular plants (*Diazgranados, 2015*; *Fernández-Alonso, 2002*; *Madriñán, Cortés & Richardson, 2013*; *Rangel-Ch, 2000*), in a short period of time (less than 4 my BP; *Cuatrecasas, 1986*; *Hooghiemstra & Van der Hammen, 2004*; *Madriñán, Cortés & Richardson, 2013*; *Van der Hammen & Cleef, 1986*). Thus, they are an ideal system to understand rapid adaptive radiations and speciation mechanisms in sky islands.

Frailejones (the name used in this work to refer to all species within subtribe Espeletiinae Cuatrec. (Asteraceae: Millerieae Lindl.)) are the most representative and iconic plants of the páramo. They are all 144+ species distributed in eight genera (Fig. 1): *Carramboa* Cuatrec. (4 spp.), *Coespeletia* Cuatrec. (8), *Espeletia* Mutis ex Humb. & Bonpl. (72), *Espeletiopsis* Cuatrec. (23), *Libanothamnus* Ernst (11), *Paramiflos* Cuatrec. (1), *Ruilopezia* Cuatrec. (24) and *Tamania* Cuatrec. (1) (*Diazgranados, 2012a*; *Diazgranados & Morillo, 2013*;
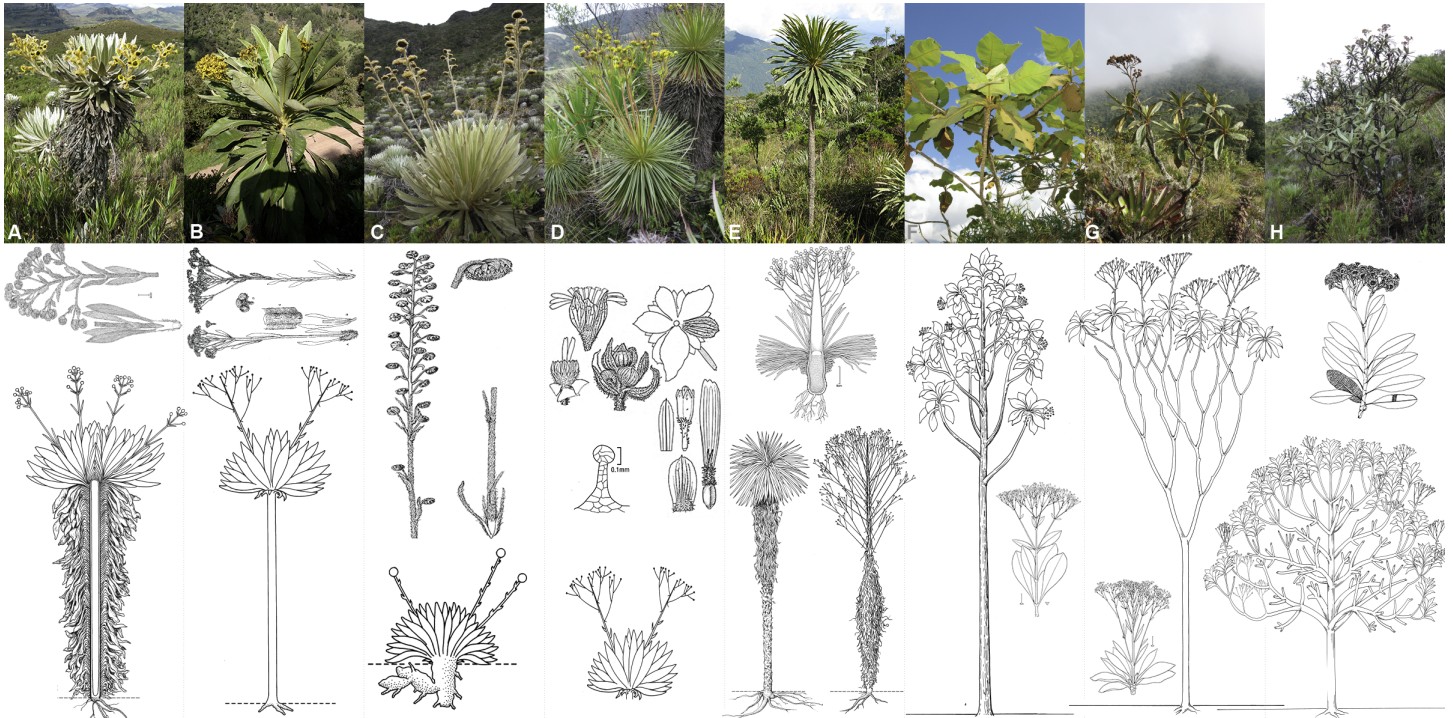

**Figure 1** **Morphological diversity in the genera of Espeletiinae.** (A) *Espeletia* (72 species); (B) *Espeletiopsis* (23); (C) *Coespeletia* (8); (D) *Paramiflos* (1); (E) *Ruilopezia* (24); (F) *Carramboa* (4); (G) *Tamania* (1); and (H) *Libanothamnus* (11). Illustrations from *Cuatrecasas (2013)*, made by Alice Tangerini and Florence Lambeth (Department of Botany, US National Herbarium, Smithsonian Institution); photographs: Diazgranados.

*Diazgranados & Sanchez, 2013*). In reality there are many undescribed taxa, for instance the first author is currently describing 14 new species discovered during the course of his field and herbarium work. Despite the interest in the frailejones, there is considerable controversy about the classification within the subtribe, and many of the genera were not resolved as monophyletic in earlier studies (*Panero, 2007*; *Rauscher, 2002*; *Sklenár et al., 2005*). The subtribe has enormous morphological variation (Fig. 1), including numerous synapomorphies: spiral leaf phyllotaxis; obpyramidal to prismatic shape of the glabrous and epappose cypselae; fertile female ray flowers and functionally male disc flowers; pluriseriate involucre and persistent pales of the receptacle; thick and woody stems; xeromorphic structure; specialized life-forms; and a static chromosome number ($n = 19$) (*Cuatrecasas, 1976*; *Cuatrecasas, 2013*; *Robinson, 1981*).

Frailejones are widely distributed and abundant in the high Andean forest and páramos of Colombia (88 spp.) and Venezuela (68 spp.); only one species occur in northern Ecuador, with an isolated population in the Sierra de Llanganates (*Diazgranados, 2012a*; *Diazgranados, 2013*). Although there are some species endemic to the high Andean forests, and a few that can succeed at altitudes as low as 1,300 m or as high as 4,780 m, most of the taxa (104 spp.) are found between 3,200–3,400 m of altitude (*Diazgranados, 2012a*). There are three apparent centers of species diversity: Mérida (with 44 spp.) in Venezuela, Santander–Norte de Santander (41 spp. combined) and Boyacá (45 spp.) in Colombia (*Cuatrecasas, 1986*; *Diazgranados, 2012a*).

Local endemism at the specific level is extremely high (ca. 90%), possibly as a result of island-like radiation on a continental scale (*Hughes & Eastwood, 2006*). Most of the species are gregarious and often represent more than 40% of plant cover (*Luteyn, 1999*; *Rangel-Ch, 2000*). Seeds lack a pappus or other disseminating device and are dispersed mainly by gravity. According to *Cuatrecasas (2013)*, wind or animals may disperse the cypselae short distances but never more than 1–3 m from the parent plant, although light rains and small streams may disperse them longer distances. Bees (principally species of *Bombus*, *Colletes* and *Apis*) are the most frequent pollinators of Espeletiinae and no long-distance pollinators are known (*Berry & Calvo, 1989*; *Berry & Calvo, 1994*; *Fagua & Gonzalez, 2007*; *Sobrevila, 1988*). Hence, both pollination and seed dispersal suggest that there is a strong isolation by distance among different páramos, which are normally separated by several kilometers of areas unsuitable for frailejones species.

The subtribe has been recently circumscribed within the tribe Millerieae Lindl., as part of the Heliantheae Alliance (*Baldwin, 2009*; *Baldwin, Wessa & Panero, 2002*; *Panero, 2007*). There is no agreement regarding Cuatrecasas' classification within the subtribe. The monophyly of the subtribe was confirmed based on nrDNA ITS with 14 frailejones species and 51 outgroups (*Rauscher, 2002*). According to this work the closest relatives are *Rumfordia* DC., *Ichthyothere* Mart. and *Smallanthus* Mack.

Speciation of the group started very recently, most likely during the late Pliocene or early Pleistocene (2–4 my BP) (*Cuatrecasas, 1986*; *Hooghiemstra & Van der Hammen, 2004*; *Torres et al., 2013*; *Van der Hammen & Cleef, 1986*), and it is likely an ongoing process. Due to this remarkable diversity, which appears to have evolved over a relatively short period of time, the group has been considered a classic example of rapid radiation in the tropics (*Cuatrecasas, 1986*; *Monasterio & Sarmiento, 1991*; *Rauscher, 2002*), although this hypothesis has never been rigorously tested. Expansions (with reconnection) and contractions (with isolation) of the páramo ecosystem during Pleistocene glaciations and inter-glaciations could have played a major role in the radiation and dispersion of these taxa (*Cuatrecasas, 2013*).

Two general hypotheses for the radiation of frailejones can be proposed:

(1) Upward migration → horizontal migration/expansion → speciation
   Ancestors colonized the high altitudes of the mountains finding available niches and subsequently expanded their distributions occupying different areas of their fundamental ecological niche. In this scenario, distant high-altitude species would be more closely related to each other than to geographically closer lower altitude species and overall morphology would reflect synapomorphies rather than convergence.

(2) Horizontal migration/expansion → upward migration → speciation
   Ancestors of the different genera migrated horizontally (i.e., at the same altitude), probably during the Pleistocene glaciations, followed by vertical migrations (upward in altitude) with subsequent allopatric speciation events. In this scenario, phylogenetic clustering (with closely distributed species more related to each other) would be more common; morphology with frequent homoplasy would be the dominant mode of evolution.

Of course it is possible that it was not one or the other but rather a combination of the two previous hypotheses. With more than 22–24 known Pleistocene glaciations and inter-glaciations it is plausible that radiation could have followed this pattern, with downward migration, hybridization and introgression during the glaciations, and upward migration and allopatric speciation during the inter-glaciations.

A well-resolved phylogeny can provide information to test these hypotheses. Although there have been some previous attempts to reconstruct the evolutionary history of this group based on morphological (*Cuatrecasas, 1976*; *Cuatrecasas, 2013*) and molecular data (*Rauscher, 2002*; *Rauscher, 2000*; *Sánchez, 2005*), relationships among genera and species remain largely unresolved. We present here the most complete phylogeny to date for frailejones and discuss Cuatrecasas' generic classification and the influence of geography in shaping the evolution of this group, in a first attempt to understand the processes underlying this spectacular radiation.

## MATERIALS & METHODS

### Taxon sampling

During major expeditions beginning in 2007, ca. 70 páramo locations of Colombia and Venezuela were visited to photograph, collect and geo-reference frailejones species. A total of 555 samples (MDC 3,537–4,092) from ca. 110 species were collected (Table 1), following standard protocols for herbarium and molecular analyses (voucher are in ANDES, COL, HECASA and to be distributed (*Thiers, 2012*)). Collections were made under permits No. 2698 of 09/23/2009 and No. 2 of 02/03/2010 (Ministerio de Ambiente, Colombia), and IE-126 (Venezuela, authorized by Petr Sklenář). Notable collections included 14 new taxa (two already published: *Coespeletia palustris* Diazgr. & Morillo and *Espeletiopsis diazii* Diazgr. & L.R.Sánchez), the first report of the genus *Ruilopezia* for Colombia (with *R. cardonae* (Cuatrec.) Cuatrec.), several new reports for localities, numerous putative hybrids, a few species previously thought to be extinct, and the identification of several critically endangered species (*Diazgranados, 2012a*; *Diazgranados, 2013*; *Diazgranados, 2015*; *Diazgranados & Morillo, 2013*; *Diazgranados & Sanchez, 2013*). Additional material was obtained from specimens at MO, US and F. *Espeletia pycnophylla* Cuatrec., the sole species found in Ecuador, was collected in the south of Colombia near the border with Ecuador, eliminating the need for field work in Ecuador.

### DNA purification

DNA extraction from frailejones is particularly complex because of the abundant indumentum of the leaves and the high concentration of terpenes and other metabolites. Extractions from pubescent tissue yield degraded DNA. Adult leaves with old indumentum are sometimes contaminated by fungi. Therefore, young developing leaves from the center of the rosette were use for this purpose, shaving the tissue part of interest. We highly recommend shaving the leaves *in situ*. An initial set of 16 species was used for high quality large scale extractions with the CTAB protocol (*Doyle & Doyle, 1987*) followed by purification via cesium chloride gradients. Subsequently the DNeasy Plant Mini Kit (Qiagen, Valencia, California, USA) was used, optimizing the Qiagen protocol (*2006*)
**Table 1  Number of species collected and documented during fieldwork for this project.** In addition to these collections, samples from other species were obtained from different sources.

| Genus | No. of species | No. collected (%)[a] | New species[b] |
|---|---|---|---|
| *Carramboa* | 4 | 3 (75) | 1 |
| *Coespeletia* | 8 | 6 (86) | 1 |
| *Espeletia* | 72 | 56 (79) | 6 |
| *Espeletiopsis* | 23 | 19 (86) | 4 |
| *Libanothamnus* | 11 | 9 (82) | 1 |
| *Paramiflos* | 1 | 1 (100) | 0 |
| *Ruilopezia* | 24 | 15 (63) | 0 |
| *Tamania* | 1 | 1 (100) | 1 |
| **TOTAL** | **144** | **110 (78)** | **14** |

**Notes.**
[a] Putative new species not counted in this column.
[b] New species being described.

to obtain comparable quality. Modifications to the manufacturer's protocol included the following: (1) after shaving and grinding the leaf fragments under liquid nitrogen, ground tissue is incubated for 24 h with 400 µl of buffer AP1, 4 µl of RNase, 60 µl of 2-$\beta$-mercaptoethanol, 60 µl of proteinase-k and 5 µl of sodium dodecyl sulfate (SDS) at 20% (w/v), inverting the tubes a few times during incubation; (2) after addition of buffer AP2, the mix is incubated on ice for one hour; and (3) when indument was not totally removed, a double amount of reagents was used. After checking for DNA quality and protein content through spectrophotometry, extracted DNA was cleaned using the following procedure: (1) incubation for 30 min at 37 °C with 0.1% (w/v) SDS, 20 µl/ml of proteinase-k; (2) precipitation by incubation for 30 min at −20 °C with 5 µl 3M of NaAc and 100 µl of EtOH 95%; (3) centrifugation at 14,000 rpm for 15 min; (4) removal of the upper phase by inversion and lyophilization; and (5) resuspension with 50 µl of AE buffer. The ingroup included DNA extracted from 240 samples, representing all eight genera and 140 species (including various new taxa being described, and *Tamananthus crinitus* V.M.Badillo); the outgroup comprised 40 samples from 26 species of the following genera: *Ichthyothere*, *Polymnia*, *Rumfordia* and *Smallanthus*. Available Genbank sequences (14 ingroup and 51 outgroup sequences for ITS generated by *Rauscher, 2002*) were compared to the sequences obtained here.

## PCR amplifications for sequence data

A combined approach of using DNA sequence data from three lines of evidence (cpDNA, nrDNA and AFLPs) was used in this project. Preliminary screening of chloroplast and nuclear regions was performed with 24 samples from species covering the entire geographic range of the subtribe. *Sánchez (2005)* explored 18 chloroplast regions based on Shaw's recommendations (2005; 1998) for 24 frailejones species, but none of the amplified regions was of utility to establish well-supported relationships at generic and species levels. We screened seven additional regions reported as highly variable (*Shaw et al., 2005*; *Shaw et al., 2007*; *Timme et al., 2007*): rpoB-trnY, ndhC-trnV, trnL-rpl32, rpl16, rps16, ycf6-psbM and trnG. The variable region trnS–trnG has an inversion in Asteraceae that prevents

amplification (*Jansen & Palmer, 1987*). Initial screenings showed rpl16 as promising, with 29 haplotypes in 47 sequences. Therefore this region was amplified for phylogenetic reconstructions, using the primers by *Small et al. (1998)*: F71 (5′-GCTATGCTTAGTGT GTGACTCGTTG-3′) and R1516 (5′-CCCTTCATTCTTCCTCTATGTTG-3′).

The nuclear ribosomal internal transcribed spacer (nrITS) has been used widely to reconstruct phylogenies in several groups of Asteraceae (*Blöch, 2010*; *Friar et al., 2008*; *Gruenstaeudl et al., 2009*; *Keeley, Forsman & Chan, 2007*; *Morgan, Korn & Mugleston, 2009*; *Schilling & Panero, 2011*; *Vaezi & Brouillet, 2009*; *Zhang et al., 2011*; and others). *Rauscher (2000)* conducted extensive work in the Espeletiinae with nrITS, using 169 accessions from 15 ingroup and 51 outgroup species. He found a level of variation between 0–4.5%, with no detected intraspecific variation in numerous species. He cloned all accessions and found that 20% of these had only one polymorphic nucleotide position (two haplotypes), while 25% had nucleotide polymorphisms at more sites, suggesting a certain level of within-individual variation. However, he reported that variants typically coalesced within the population and/or species, with no effect on phylogenetic inference; furthermore, several species yielded exactly the same sequences. To build upon Rauscher's work, ITS1–5.8S–ITS2 (ITS, hereafter) was also used, with the universal primers: ITS-1AF (5′-TCCTTCCGCTT ATTGATATGC-3′) and ITS-4R (5′-GGAAGTAAAAGTCGTAACAAGG-3′) (*White et al., 1990*). We tested for the presence of pseudogenic ITS regions as a consequence of incomplete concerted evolution based on the identification of the conserved 5.8S motifs (CGATGAAGAACGTAGC, GAATTGCAGAATCC and TTTGAACGCA) and a GC% comparison across all the sequences (*Harpke & Peterson, 2008a*; *Harpke & Peterson, 2008b*). No pseudogenic copies were found.

Additional nuclear DNA regions, including single copy genes, were screened without success, either because of very low variability (e.g., gapC), high complexity and multiple bands (e.g., *leafy*, *pepC*), or unsuccessful amplifications (e.g., *waxy*). However, the external transcribed spacer region (ETS) of the nuclear 18S-26S ribosomal repeat showed fairly good phylogenetic resolution power. Numerous studies in related groups have used ETS for reconstructing phylogenies (*Baldwin & Markos, 1998*; *Clevinger & Panero, 2000*; *Ekenäs, Baldwin & Andreasen, 2007*; *Garcia et al., 2011*; *Masuda, Yukawa & Kondo, 2009*; *Mavrodiev et al., 2008*; *Moore et al., 2012*; *Morgan, Korn & Mugleston, 2009*; *Schilling & Panero, 2011*; *Soltis et al., 2008*; *Timme, Simpson & Linder, 2007*; *Wahrmund et al., 2010*; and others). *Timme, Simpson & Linder (2007)* reported a large region of one to five subrepeats (each of ∼250 bp) in the ETS for *Helianthus* L., with intraspecific variation, evidenced by multiple bands during gel electrophoresis. Although we did not find multiple bands in frailejones, we did find interspecific variation in the number of subrepeats. The length of this region in Espeletiinae is between ∼1,450 and 2,500 bp. Therefore, in addition to external primers, internal primers were used: ETS1f (5′-CTTTTTGTGCA TAATGTATATATAGGGGG-3′), ETS2f (5′-CTGAGCCCCACTTCGGTAGTTTGGC-3′), 11r (5′-CAAACCAAACACCACTCATGCACC-3′), and 18S2l (5′-TGACTACTGGCAGGA TCAACCAG-3′) (*Timme, Simpson & Linder, 2007*). Three additional internal primers were developed for this study: ETS3f (5′-GASCTGACGAAGTACCCATGA-3′), ETS4f (5′-CTCAATGGGCCACAACATC-3′) and 10r (5′-CGGGTGGCTAATTGTTGG-3′).

All polymerase chain reactions (PCR) were carried out in a final volume of 25 μl. Reactions were prepared with reagents from the GoTaq DNA polymerase kit (Promega, Madison, WI, USA), with 0.5 μl of template DNA (∼10–100 ng), 5 μl of 5X Green GoTaq Reaction Buffer, 2.5 μl of MgCl$_2$ (25 Mm), 1.5 μl dNTPs (10 mM), 0.5 μl of each primer (10 mM), 0.25 μl of GoTaq DNA polymerase, 1.5 μl of DMSO and 0.5 μl of bovine serum albumin (BSA; 0.1 μg/μl). Amplifications were performed using Eppendorf (Westbury, New York, USA) Mastercycler gradient ep thermal cyclers. Cycler programs were: for rpl16: denaturation at 94 °C for 2 min, 30 cycles of denaturation at 95 °C for 1 min, annealing at 52 °C for 1 min, and extension at 65 °C for 4 min, followed by a final extension at 65 °C for 10 min; for ITS: denaturation at 94 °C for 2 min, 35 cycles of denaturation at 94 °C for 20 s, annealing at 54 °C for 35 s, and extension at 72 °C for 45 s, followed by a final extension at 72 °C for 7 min; and for ETS: denaturation at 94 °C for 5 min, 30 cycles of denaturation at 94 °C for 30 s, annealing at 50 °C for 30 s, and extension at 72 °C for 1.5 min, followed by a final extension at 72 °C for 5 min PCR products were checked on 1% agarose gels before being cleaned with the QIAquick PCR purification kit (Qiagen, Valencia, California, USA). PCR products were sent for sequencing to Macrogen Inc. (Seoul, South Korea). Consensus sequences were assembled in Sequencher$^{TM}$ 4.2, and aligned manually using Se-Al v2.0a11 (*Rambaut, 1996*). Concatenation of matrices and final editing of nexus files were conducted in Mesquite 2.75 (*Maddison & Maddison, 2011*). Sequences are available in GenBank, with accession numbers: ETS: KY231675–KY231818, ITS: KY231383–KY231532 and rpl16: KY231533–KY231674. GenBank accession numbers and voucher information per sample are provided in Table S1.

## AFLP amplifications and genotyping

In addition to sequence data, amplified fragment-length polymorphisms (AFLPs) were used to infer relationships among the species, and particularly to resolve polytomies or nodes with low or no support by sequence data. AFLPs are variable markers typically used for fingerprinting, estimating relatedness, genetic mapping and more recently for reconstructing phylogenies (*Avise, 2004*; *McKinnon et al., 2008*; *Meudt & Clarke, 2007*; *Vos et al., 1995*). Numerous phylogenetic studies in recently radiated groups have used AFLPs (*Jabaily, 2009*; *Koopman et al., 2008*; *McKinnon et al., 2008*; *Schmidt-Lebuhn, 2007*; *Schmidt-Lebuhn, Kessler & Kumar, 2009*; *Worley, Ghazvini & Schemske, 2009*). These markers have also been used successfully in phylogenetic reconstructions in Asteraceae (*Koopman, 2005*; *Koopman, Zevenbergen & Van den Berg, 2001*). Acquisition of AFLP data followed a modified version of the protocol by *Trybush et al. (2006)*. DNA was digested for 1 h at 37 °C in 30 μL of total volume with 0.25 μl of EcoRI, 0.5 μl of MseI, 3 μl of 1x NEB buffer 4 (New England Biolabs, Ipswich, MA), 1.5 μg of BSA and 2 μl of template DNA. Each pair of EcoRI and MseI adaptors was annealed after incubation for 10 min at 65 °C, and allowed to cool very slowly for 4 h. Ligation included the 30 μl digestion mix plus a 10 μl mix containing 1.5 μl of T4 DNA ligase (60 U), 1 μl of ATP, 1 μl of 1x NEB buffer 4, 0.5 μl of the annealed EcoRI adaptors, 5.0 μl of the annealed MseI adaptors and 1 μl of ddH$_2$O. The ligation reaction was incubated at 37 °C for 4 h 15 min. The digested-ligated mix was 10-fold diluted. Pre-amplifications were prepared in a total volume of 10 μl, with

2 µl of 5X Green GoTaq Reaction Buffer, 1 µl of $MgCl_2$, 0.25 µl of dNTPs (10 mM), 0.05 µl of each pre-selective primer (*Eco*RI primer + A and *Mse*I primer + C), 0.6 µl of DMSO, 0.2 µl of BSA 1X, 2.5 µl of the digested-ligated template DNA, 0.1 µl of GoTaq DNA polymerase and 2,35 µl of $ddH_2O$. Thermocycling was performed with a first step of enzyme denaturation at 65 °C for 5 min, followed by 30 cycles of 30 s at 94 °C, 30 s at 56 °C and 1 min at 72 °C. PCR products with pre-amplified DNA were 20-fold diluted. Selective amplifications were performed in 10 µl reactions with the same quantities and reagents as for the pre-selective PCR, except for the addition of 0.07 µl of each selective primer (*Eco*RI primer + 3b and *Mse*I primer + 3b) and only 1 µl of pre-amplified DNA. Selective reactions included denaturation at 95 °C for 15 min, 13 cycles of 30 s at 94 °C, 1 min at 65 °C with a ramp temperature of –0.7 °C/cycle, and 1 min at 72 °C; then 25 cycles of 30 s at 94 °C, 30 s at 55 °C and 1 min at 72 °C; and a final extension of 10 min at 72 °C.

WellRED D2-, D3- and D4-PA fluorescent primers (Sigma-Aldrich, St. Louis, MO) were used for selective amplifications. In total, 36 primer combinations were tested following recommendations for *Helianthus* (*Applied Biosystems, 2007*). Five combinations failed to amplify (selective 3b for EcoRI-selective 3b for MseI: ACT-CAC, ACC-CAG, ACT-CAG, AAC-CAT and ACT-CAT), and three primer combinations showed only weak amplification (ACG-CAA, ACG-CAC and ACT-CAA). The remaining 28 primer combinations were used for genotyping (Table 2). Results from each step were checked by electrophoresis on 1.5% agarose gels. Each 96-well plate contained 81 samples, of which 12 (∼15%) were replicated. In addition, 12 samples were repeated between plates and where possible, multiple samples were included per species. Genotyping was perfomed on a Beckman-Coulter CEQ8000 sequencer, multiplexing the reactions with the three WellRED dyes, in a final volume of 10 µl. Scoring was conservative, with only the strongest allele peaks scored. Alleles not consistent between replicated samples or between different runs for the same sample were discarded. Minimum bin (peak) width was set to 1 base (b). Bins of <60 b were eliminated. To determine minimum fragment size and minimum intensity threshold, multiple partitions were tested by measuring homoplasy and other tree descriptors (see Table 3). Since a preponderance of phylogenetic signal was found in sizes between 60 and 100 b, the minimum size of 60 b was retained. The minimum intensity threshold with phylogenetic signal was established at 1,000 relative fluorescent units (RFU). From an initial total of 5,551 fragments, only 1,665 were retained. Samples with poor amplifications were discarded. In total, 118 ingroup and 20 outgroup taxa were genotyped.

**Phylogenetic reconstruction**

Phylogenies based on sequence data were reconstructed under maximum parsimony (MP), maximum likelihood (ML) and Bayesian inference (BI). The AFLP dataset employed minimum evolution (ME) analyses. Trees were rooted with *Smallanthus*, *Ichthyothere* and *Rumfordia* species, based on *Rauscher (2002)*. MP analyses were performed in Paup* version 4.0b10 (*Swofford, 2002*). For all analyses, gaps were treated as missing data and ambiguous positions were excluded. Full heuristic searches were configured with MAXTREES set to 100,000, with the tree-bisection-and-reconnection (TBR) branch swapping algorithm, 10 random additions, and holding 1 tree at each step. Non-parametric

**Table 2 AFLP primer combinations used for genotyping.** First triplet for each primer combination corresponds to the selective anchor for EcoRI; second triplet corresponds to the selective anchor for MseI. Mean Y threshold corresponds to the intensity of each band (peak height), in relative fluorescence units (RFU).

| Primer combination | Number of fragments | Minimum size (b) | Maximum size (b) | Mean size (b) | Variance of size (b$^2$) | Mean Y threshold (RFU) |
|---|---|---|---|---|---|---|
| AAC-CAA | 39 | 61.7 | 417.5 | 135.5 | 0.0 | 4603.9 |
| AAC-CAC | 43 | 65.9 | 231.5 | 130.0 | 0.1 | 4296.3 |
| AAC-CAG | 41 | 61.7 | 348.5 | 116.2 | 0.0 | 9722.0 |
| AAC-CAT | 51 | 59.8 | 376.6 | 141.9 | 0.0 | 4027.3 |
| AAG-CAA | 39 | 62.5 | 294.0 | 144.6 | 0.0 | 14775.7 |
| AAG-CAC | 56 | 59.6 | 417.8 | 160.3 | 0.1 | 9138.6 |
| AAG-CAG | 39 | 59.8 | 259.3 | 158.6 | 0.1 | 10322.9 |
| AAG-CAT | 69 | 60.0 | 311.0 | 128.5 | 0.1 | 7942.1 |
| ACA-CAA | 47 | 63.1 | 307.0 | 155.7 | 0.0 | 8447.5 |
| ACA-CAC | 65 | 64.9 | 310.4 | 160.3 | 0.1 | 8102.3 |
| ACA-CAG | 58 | 63.8 | 334.5 | 159.8 | 0.1 | 10157.2 |
| ACA-CAT | 62 | 63.5 | 351.1 | 170.6 | 0.0 | 4192.8 |
| ACG-CAA | 56 | 66.7 | 412.5 | 174.5 | 0.0 | 4644.9 |
| ACG-CAC | 51 | 60.4 | 267.0 | 139.4 | 0.0 | 6245.9 |
| ACG-CAG | 58 | 59.5 | 307.2 | 148.4 | 0.0 | 6970.9 |
| ACG-CAT | 51 | 64.9 | 348.0 | 158.3 | 0.1 | 3253.5 |
| ACT-CAA | 91 | 62.6 | 443.8 | 169.9 | 0.0 | 15551.5 |
| ACT-CAC | 59 | 60.7 | 383.9 | 219.0 | 0.1 | 8792.9 |
| ACT-CAG | 58 | 65.9 | 378.5 | 181.1 | 0.1 | 11644.2 |
| ACT-CAT | 83 | 59.0 | 360.8 | 201.9 | 0.1 | 10579.9 |
| AGC-CAA | 54 | 61.4 | 318.2 | 164.8 | 0.0 | 3303.6 |
| AGC-CAC | 40 | 59.2 | 281.8 | 137.8 | 0.1 | 5745.0 |
| AGC-CAG | 57 | 62.9 | 363.1 | 143.6 | 0.0 | 7534.2 |
| AGC-CAT | 40 | 62.0 | 320.3 | 158.4 | 0.0 | 3997.4 |
| AGG-CAA | 75 | 58.2 | 360.5 | 182.6 | 0.0 | 10933.4 |
| AGG-CAC | 109 | 59.6 | 436.4 | 185.5 | 0.2 | 12275.0 |
| AGG-CAG | 86 | 63.8 | 381.6 | 190.8 | 0.1 | 16182.5 |
| AGG-CAT | 88 | 59.7 | 369.7 | 193.8 | 0.1 | 9577.2 |
| **Total** | **1,665** | **58.2** | **443.8** | **165.5** | **0.1** | **8892.5** |

boostrap searches were estimated with 100,000 replicates and MAXTREES set to 1. For ML and BI reconstructions, evolutionary models were selected using the AIC criterion in JModelTest 0.1.1 (*Posada, 2008*), which uses the Phyml algorithm to test 88 models in large phylogenies by maximum likelihood (*Guindon, 2003*). ML reconstructions were performed in Garli 2.0.1019 (*Zwickl, 2006*) at the the CIPRES Science Gateway V. 3.1 (http://www.phylo.org/sub_sections/portal/). The candidate model of evolution selected via AIC by JModelTest was established for the ML searches without optimization, allowing Garli to infer parameter values for each model. Each data set was analyzed in eight independent runs. Boostrap analyses were performed with 120 replicates and 8 independent runs. The CIPRES portal allows a maximum of 100 replications × runs, therefore each

**Table 3 Sequence characteristics and tree statistics for all data sets.** Tree scores were estimated from the MP consensus tree.

| Statistics | rpl16 | ETS | ITS | AFLPs | ETS-ITS | ETS-ITS-rpl16 | All combined |
|---|---|---|---|---|---|---|---|
| Aligned length (characters) | 962 | 1,324 | 649 | 1,665 | 1,973 | 2,935 | 4,600 |
| Number of samples | 144[a] | 150[b] | 142[c] | 134 | 149 | 145 | 111 |
| Number of species | 127 | 119 | 130 | 115 | 117 | 116 | 95 |
| Variable characters | 93 | 560 | 245 | 1,665 | 750 | 831 | 2,294 |
| Informative characters | 35 | 368 | 173 | 1,583 | 507 | 551 | 1,818 |
| Number of MP trees | 100,000 | 100,000 | 100,000 | 20 | 100,000 | 100,000 | 24 |
| Tree length (best MP tree) | 84 | 1,155 | 527 | 10,200 | 1,545 | 1,760 | 9,797 |
| CI[*] | 0.464 | 0.113 | 0.454 | 0.159 | 0.338 | 0.405 | 0.200 |
| RI[*] | 0.930 | 0.061 | 0.841 | 0.159 | 0.747 | 0.813 | 0.456 |
| RC[*] | 0.432 | 0.007 | 0.381 | 0.057 | 0.252 | 0.329 | 0.091 |
| HI[*] | 0.536 | 0.887 | 0.546 | 0.841 | 0.662 | 0.595 | 0.800 |
| Model of evolution | TVM + I + G | TrN + G | TIM2 + G | Mkv | N/A | N/A | N/A |
| $-\ln L$ | $-2288.46$ | $-9409.64$ | $-3944.67$ | $-44743.38$ | $-14066.11$ | $-17054.85$ | $-54659.98$ |

**Notes.**
[a] Genbank accession numbers KY231675–KY231818.
[b] Genbank accession numbers KY231383–KY231532.
[c] Genbank accession numbers KY231533–KY231674.
[*] Statistics based on one of the shortest maximum parsimony trees: CI, consistency index; RI, retention index; RC, rescaled consistency index; HI: homoplasy index.

bootstrap analysis was performed in 10 tasks of 12 × 8. Bayesian inference was obtained in MrBayes v3.1.2 implemented also at the CIPRES portal. As with ML analyses, models of evolution were tested for each partition. Settings included 2 runs, 8 independent chains, temperature factor of 0.05, 50,000,000 generations sampling every 5,000 trees, and discarding the first 8,000 trees as burn-in. Data sets containing AFLPs were run for 150,000,000 generations with a temperature factor of 0.005 and a stopping rule when convergence reached 0.01, sampling every 5,000 trees and discarding the first 17,500 trees. ME analyses were performed in PAUP* with the Nei-Li distance (fragments), a starting tree by neighbor joining, TBR branch swapping and MulTrees on. Bootstrap searches had 10,000 replicates. Partitions (ETS, ITS, rpl16 and AFLPs) were analyzed both separately and combined. One-tailed Shimodaira–Hasegawa (SH) tests (*Shimodaira & Hasegawa, 1999*) were conducted in PAUP* to compare between topologies, and incongruence length difference (ILD) tests (*Farris et al., 1995*), implemented as partition homogeneity in PAUP*, were used to assess combinability of partitions. The ILD test was run with 100 replications and 10 random additions. Bootstrap support values and posterior probabilities were summarized with the function Sumtrees implemented in DendroPy (*Sukumaran & Holder, 2010*). Concatenation of support values was performed with the function Sumlabels in the same program. FigTree v1.3.1 (*Rambaut, 2009*) was used to edit the trees. A Monte Carlo permutation test with 1,000 permutations, as implemented in GenGIS (*Parks et al., 2009*), was used to test for the influence of the geography on the phylogeny. Two additional analyses were run to further verify the resulting topologies: an unrooted phylogenetic network (split network) on the ETS-ITS-rpl16 data set, with the NeighborNet algorithm as implemented in SplitsTree4 (*Huson & Bryant, 2006*); and a haplotype network based on rpl16, with equal weighting on transversions/transitions, the *median-joining* (MJ) network

algorithm, and "Connection cost" distance method, built with Network 4.6.1.1. (*Fluxus Technology Ltd, 2012*). The split network displays simultaneously all the possible networks, without giving any hierarchy. Haplotype networks, despite phylogenetic approaches, use autapomorphic characters (unique mutations) for the calculations.

## RESULTS

Field collections included ∼75% of the recognized taxa. During the development of this project 17 new species were published by other researchers (*Díaz-Piedrahita & Rodriguez-Cabeza, 2008*; *Díaz-Piedrahita & Rodriguez-Cabeza, 2010*; *Díaz-Piedrahita & Rodriguez-Cabeza, 2011*; *Díaz-Piedrahita, Rodriguez-Cabeza & Galindo-Tarazona, 2006*; *Diazgranados & Morillo, 2013*; *Diazgranados & Sanchez, 2013*). Most of these newer taxa remain unsampled in the field by us, and fragments from herbarium specimens could not be amplified due to the preservation techniques used when they were collected. A few species known from only a single collection have not been found in decades (e.g., *Espeletia canescens* A.C.Sm., *E. marnixiana* S.Díaz & Pedraza, *E. miradorensis* (Cuatrec.) Cuatrec., *E. tapirophila* Cuatrec., *E. tillettii* Cuatrec., *Espeletiopsis trianae* (Cuatrec.) Cuatrec., *Ruilopezia usubillagae* Cuatrec., etc.); furthermore, those species were collected in areas that are currently politically unstable or with very difficult access. *Tamananthus crinitus*, included in the subtribe by *Panero (2007)*, is known from only one poorly preserved specimen, glued to the herbarium sheet (US). DNA amplifications for *T. crinitus* were unsuccessful, but amplifications for AFLPs were fruitful. Two recognized hybrids (*Diazgranados, 2012a*) with clean sequences were included in most of the data sets: *Espeletiopsis × bogotensis* (Cuatrec.) Cuatrec. (= *Espeletiopsis corymbosa* Bonpl. × *Espeletia grandiflora* Bonpl.), and *Espeletiopsis × cristalinensis* (Cuatrec.) Cuatrec.

Success in DNA purification varied from taxon to taxon. In general, we found that extractions are particularly difficult from species with leaves that are highly tomentose (e.g., *Espeletia paipana* S. Díaz & Pedraza) or extremely coriaceous (*Espeletiopsis jimenez-quesadae* (Cuatrec.) Cuatrec.). Sequences were not successfully obtained from all samples for all markers. Therefore, combined data sets vary in the number of the species to secure the maximum possible coverage. Statistics for all data sets are shown in Table 3 and sequences used for reconstructions are shown in Table S1. The chloroplast region showed very little interspecific and intergeneric variation within the subtribe. Only 35 (3.6%) characters were informative, including outgroups. Within the subtribe (after excluding uninformative characters), many species even from different genera share exactly the same sequence in the alignment. Trees from rpl16 (not shown) had little resolution, with only a few clades recovered in the strict consensus tree: one clade including the Venezuelan species and one clade comprising the Colombian species (including *Espeletia pycnophylla*, present in both Colombia and Ecuador).

Most ITS sequences were clean and easily readable, showing only one band in gels and none or only a few double peaks; these positions were coded using IUPAC notations. Noisy sequences were re-amplified, and subsequently discarded if the problem persisted. For our data sets, including outgroups (13 species), 173 (26.7%) characters were informative;

excluding outgroups, only 90 (13.9%) characters were informative. The ingroup for ITS included 119 species of frailejones (84.4%). Resolution based on ITS was better than rpl16 but still insufficient to resolve the phylogeny (tree not shown). The subtribe was recovered as monophyletic, in agreement with *Rauscher (2002)*. Clades comprising Colombian and Venezuelan species were also recovered, but with low support. In the ML tree the following genera were recovered in clades with low support: *Carramboa* species, *Libanothamnus* species including *E. × cristalinensis*, most of the *Ruilopezia* species including the monotypic *Tamania*, and two clades of Colombian *Espeletiopsis* nested within a big clade of Colombian *Espeletia*.

ETS proved to be much more informative than ITS. However, amplification was difficult due to the internal repeats. Outgroups have a very large indel (∼700 bp) of repeats in the 5′ end. In *Carramboa* species, this indel is reduced to ∼560 bp, whereas in most of the remaining Venezuelan species (except for the *Libanothamnus* taxa) the indel is interrupted by four or five conserved regions of ∼28 bp, so that the large indel is replaced by 5–6 smaller indels of ∼84 bp. These indels disappear in most of the Colombian species and *Libanothamnus*. A second section (∼300 bp) downstream with two additional large indels extends from there. This second section is very difficult to amplify due to the previous repeats. Therefore, this entire section was excluded from the analyses. Sequences in the 3′ end are, by contrast, highly conserved. The complete ETS region varies from ∼1,450 to 2,500 bp. The final alignment did not include the far 3′ end because of low variability. Instead, the region between primers ETS1f and 11r (including internal primers) was used (1,324 characters in alignment). Coverage included 151 sequences comprising 119 species. As with ITS, samples with multiple bands or noisy sequences were discarded. Contig assemblage included as many as 12 single stranded sequences, and samples were often re-amplified to verify sequences. Bootstrap analyses for the ML tree for ETS produced 64 nodes with support >60%, and 25 nodes with support >90%.

The best ME tree for the AFLP data had a score of 3.39528, and the length of the shortest ME tree was 10,490 steps. The data set showed a very large amount of homoplasy (HI: 0.841), but phylogenetic signal was evident. A phylogeny based on AFLPs resolves the subtribe as monophyletic, although with support values <50 (Fig. 2). *Tamanathus crinitus*, included as part of the ingroup, is placed within the outgroup, supporting the exclusion of this genus from the subtribe (*Diazgranados, 2012a*). A number of clades have good support; however, there is no support for the backbone of the tree. An unsupported clade of four species, two from Colombia and two from Venezuela, is sister to the rest of the Espeletiinae. This result could be a product of homoplasy or of incomplete amplifications for those samples due to DNA quality. The tree shows two large reciprocally monophyletic clades with distinct geographic structure. One clade contains two smaller clades: one with primarily Venezuelan species and a few neighboring Colombian species; and other containing northern Colombian species (Fig. 2). Within the Venezuelan clade, *Carramboa badilloi* (Cuatrec.) Cuatrec. (three samples of two varieties) appears to be monophyletic. *Carramboa rodriguezii* (Cuatrec.) Cuatrec. falls in a sister clade but with no support. *Carramboa trujillensis* (Cuatrec.) Cuatrec. is grouped with a sympatric species, *Libanothamnus griffini* (Ruiz-Terán & López-Fig.) Cuatrec. with high posterior probability

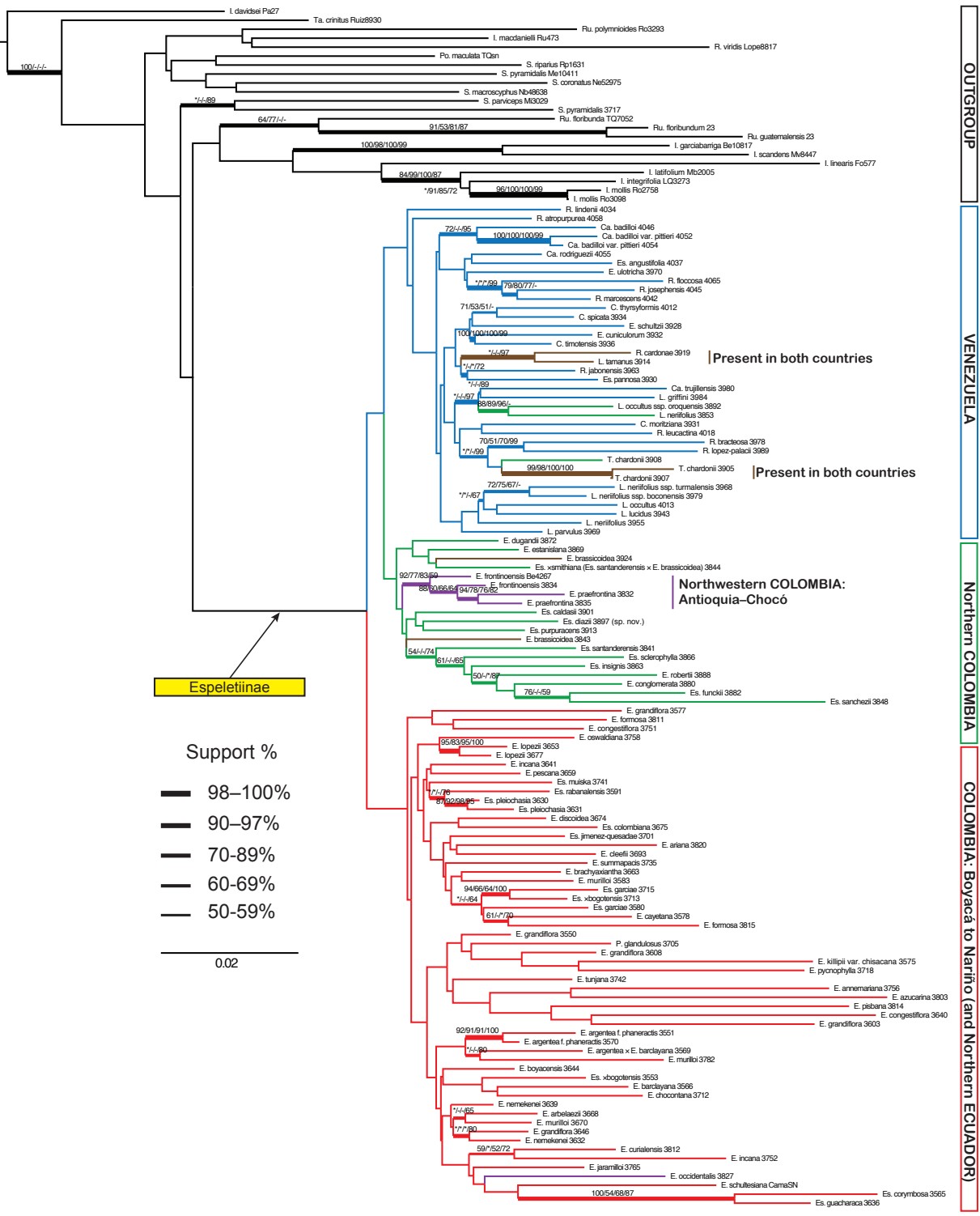

**Figure 2** **Minimum evolution (ME) tree for AFLPs.** Branch support is indicated in the following order: ME bootstrap value/ML bootstrap value/MP bootstrap value/BI posterior probability (×100); *, value below 50; -, missing value. Support values shown only for nodes with at least one support metric ≥50. Genera abbreviations: I, *Ichthyothere*; Ru, *Rumfordia*; Po., *Polymnia*; S., *Smallanthus*; Ca, *Carramboa*; C., *Coespeletia*; E., *Espeletia*; Es., *Espeletiopsis*; L., *Libanothamnus*; P., *Paramiflos*; R., *Ruilopezia*; T., *Tamania*; Ta., *Tamananthus*.

(PP). *Tamania* appears to be sister to a few *Ruilopezia* species. Most of the *Libanothamnus* species form a clade containing some nodes with fair support. The smaller clade of northern Colombian taxa comprises species restricted to Santander and Norte de Santander, two states bordering Venezuela, with the exception of two species (*Espeletia frontinoensis* Cuatrec. and *E. praefrontina* Cuatrec.) from the extreme northwest of Colombia. The second large clade within the subtribe contains only Colombian species, from Boyacá down to the limits with Ecuador. Most of the species within this clade belong to *Espeletia*, except for seven species of *Espeletiopsis* and the monotypic *Paramiflos*. Some species (e.g., *Espeletia lopezii* Cuatrec., *E. argentea* Bonpl., *Espeletiopsis pleiochasia* (Cuatrec.) Cuatrec.) show strong support for their monophyly.

The partition homogeneity test (ILD test) suggested some incongruence between the nuclear ribosomal regions ($p = 0.01$), and between nuclear ribosomal and chloroplast regions ($p = 0.14$). However, topologically these trees are almost identical, with only a few conflicting positions (Figs. 3 and 4). In both trees combining sequence data, the monophyly of the subtribe is strongly supported with a clear geographic structure, in general agreement with the phylogeny based on AFLPs. The tree based on the three regions (ETS-ITS-rpl16) showed increased support for most nodes, and more resolution than the trees based solely on nrDNA data or on individual partitions. In this tree, numerous clades are well supported. *Coespeletia* species (except for *C. moritziana* (Sch.Bip. ex Wedd.) Cuatrec.) fall in the same clade along with the sympatric *Espeletia semiglobulata* Cuatrec. and *E. cuniculorum* Cuatrec., all from the superpáramos of Mérida (Venezuela). The tree recovers a clade containing *Espeletiopsis pannosa* (Standl.) Cuatrec. and *E. angustifolia* (Cuatrec.) Cuatrec., two Venezuelan species with sericeous silvery indumentum and white-purplish flowers, and *Ruilopezia floccose* (Standl.) Cuatrec., another species with silvery indumentum. Three similar *Ruilopezia* species (*R. marcescens* (S.F.Blake) Cuatrec., *R. lindenii* (Sch.Bip. ex Wedd.) Cuatrec. and *R. leucactina* (Cuatrec.) Cuatrec.) are grouped in a clade separated from the rest of the species of the genus. *Espeletia schultzii* Wedd. forms a well-supported clade with *E. aristeguietana* Cuatrec. and *E. jajoensis* Aristeg. Another clade of very similar species is recovered: *Espeletia ulotricha* Cuatrec., *E. nana* Cuatrec. and *E. marthae* Cuatrec. The latter species are all very small plants from rocky páramos, with simplification of synflorescence, small leaves and rectangular sheaths. Most of the *Ruilopezia* species form a clade, with *Tamania* nested within. The monophyly of *Libanothamnus* is supported, although the clade contains *R. cardonae*, the southernmost species of the genus, and *Espeletiopsis × cristalinensis*. The Colombian clade is only well-supported by posterior probability. A clade with most of the species of *Espeletiopsis* is recovered. *Paramiflos* is nested within a small *Espeletiopsis* clade. Several small clades geographically meaningful are also supported.

A tree reconstructed with all the available molecular data (sequence data and AFLPs) loses some resolution, and some nodes with very low support collapse (Fig. 5). However, the base of the Espeletiinae clade is more resolved, showing two large clades: one of primarily Venezuelan species that includes a few neighboring Colombian species; and a second clade of only Colombian species, depicting a shallower resolution. Despite some loss of resolution, the geographic structure is still obvious even in the Colombian clade.

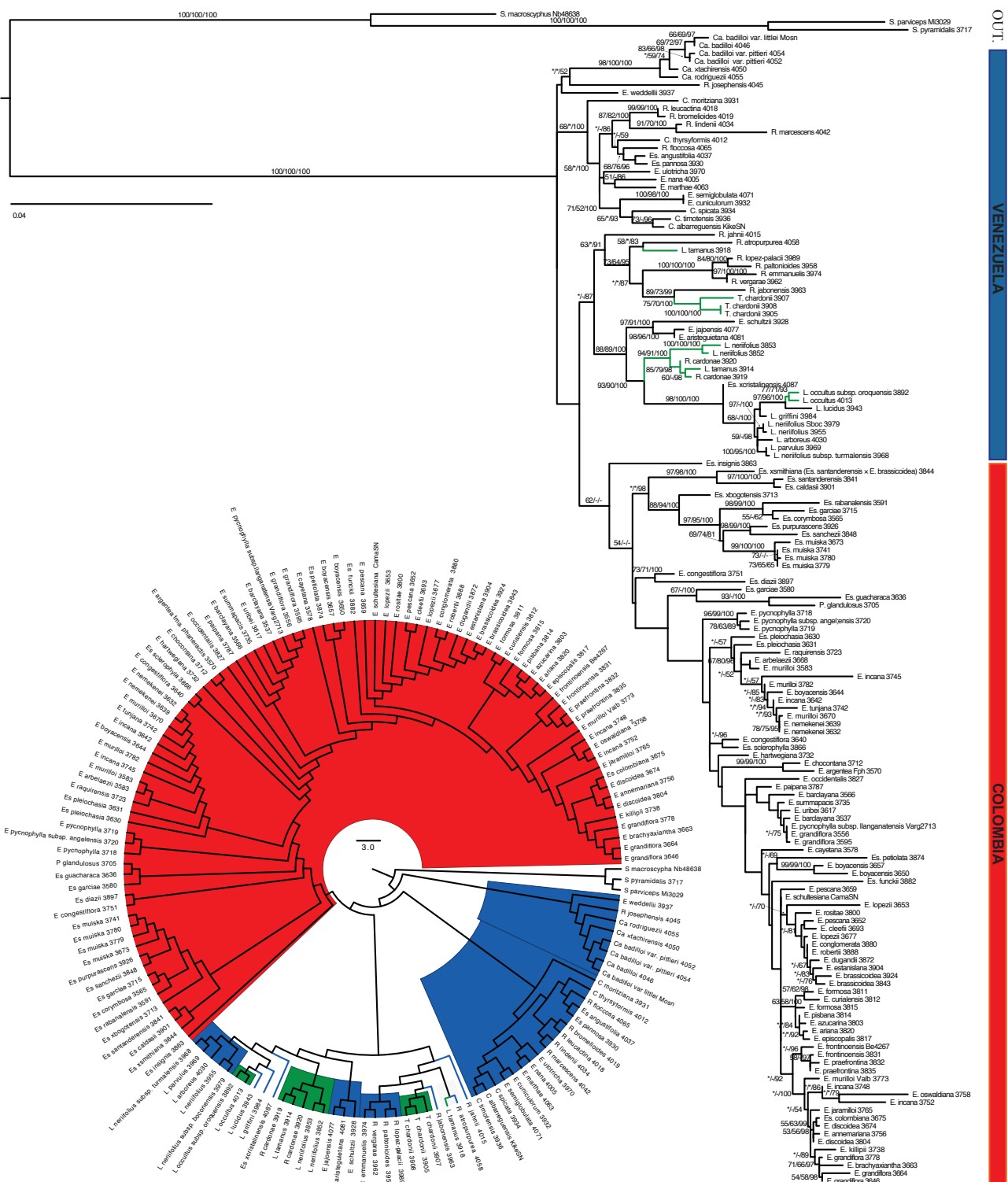

**Figure 3  Maximum likelihood tree based on ITS and ETS.** Branch support is indicated in the following order: ML bootstrap value/MP bootstrap value/BI posterior probability (×100); *, value below 50; -, missing value. Support values shown only for nodes with at least one support metric ≥50. In green: taxa present in the border between the two countries. Genera abbreviations: S., *Smallanthus*; Ca., *Carramboa*; C., *Coespeletia*; E., *Espeletia*; Es., *Espeletiopsis*; L. *Libanothamnus*; P., *Paramiflos*; R., *Ruilopezia*; T., *Tamania*.

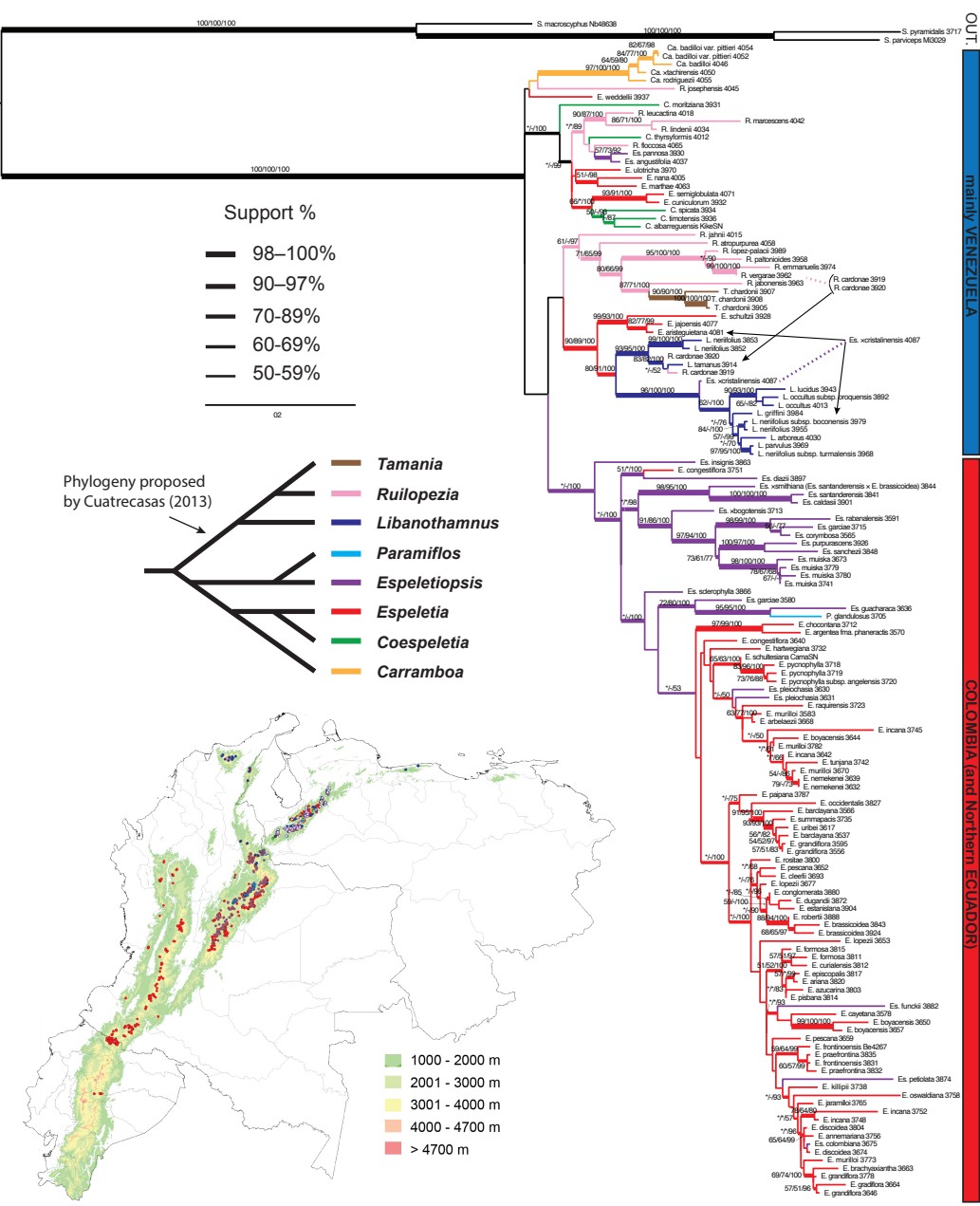

**Figure 4 Maximum likelihood tree based on ITS, ETS and rpl16.** Branch support is indicated over the branch or close to the angle of the branch in the following order: ML bootstrap value/MP bootstrap value/BI posterior probability ($\times100$); *, value below 50; -, missing value. Support values shown only for nodes with at least one support metric $\geq$50. Genera abbreviations: S., *Smallanthus*; Ca., *Carramboa*; C., *Coespeletia*; E., *Espeletia*; Es., *Espeletiopsis*; L., *Libanothamnus*; P., *Paramiflos*; R., *Ruilopezia*; T., *Tamania*. Schematic phylogeny proposed by *Cuatrecasas (2013)* displayed with black lines. Geographic distribution of genera (colored dots) shown on map. Background color denotes mountains over 1,000 m of altitude.

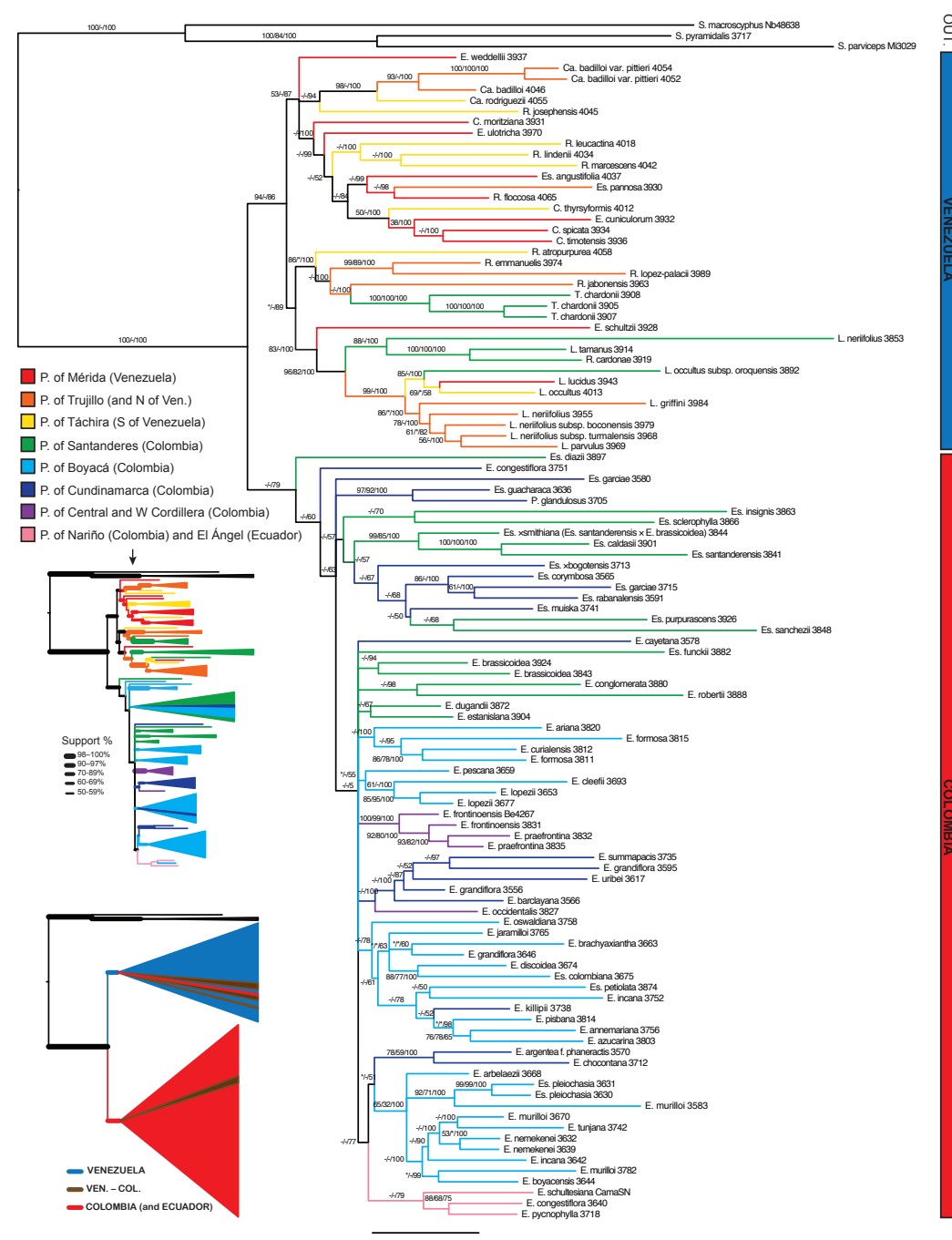

**Figure 5** **Bayesian tree estimated with all the available molecular evidence (ETS, ITS, rpl16 and AFLPs).** Branch support is indicated over the branch or close to the angle of the branch in the following order: ML bootstrap value/MP bootstrap value/BI posterior probability (×100); ⋆, value below 50; -, missing value. Support values shown only for nodes with at least one support metric ≥50. Genera abbreviations: S., *Smallanthus*; Ca., *Carramboa*; C., *Coespeletia*; E., *Espeletia*; Es., *Espeletiopsis*; L., *Libanothamnus*; P., *Paramiflos*; R., *Ruilopezia*; T., *Tamania*. Main páramo massifs indicated in colors.

Furthermore, the topology is congruent with trees generated only with sequence data, and all the small clades described previously are retained. Both the split network based on ITS-ETS-rpl16 (Fig. S1) and the haplotype network with rpl16 (Fig. S2) show this striking geographic structure. It is particularly interesting to note in the haplotype network that one cluster of haplotypes show only Colombian taxa, while the other cluster includes all the Venezuelan species, a few Colombian species and some haplotypes shared by taxa from the two countries. Lastly, a Monte Carlo permutation test performed with the ML reconstruction suggests a very strong influence of the geography on the phylogenetic relationships ($p = 0.00$, Fig. 6).

## DISCUSSION

### Molecular markers

Evolutionary relationships within numerous recent rapid radiations in animals and plants are still obscure because of the low phylogenetic signal of conventional markers. An exhaustive screening of 25 chloroplast regions for frailejones has been carried out in this work and by *Sánchez (2005)*, with unsatisfactory results. The only chloroplast region to show usable variation was rpl16 and therefore it was selected to investigate sequence variability throughout the entire subtribe. With a length of 962 bases when aligned, only 3.5% were informative characters, which was insufficient to resolve the phylogeny. ITS was more variable, but only large clades were recovered but with low support. ETS turned out to be a much more informative region. According to *Baldwin & Markos (1998)*, ETS is part of the same transcription unit as the ITS region and consequently should not be regarded as an independent line of phylogenetic evidence for comparison with ITS results. These authors affirmed that ETS can fulfill the need for additional nucleotide characters to augment the phylogenetic signal of ITS in young angiosperm clades.

The partition homogeneity test (ILD test) suggested some levels of incongruence between these ITS and ETS, and between the nuclear ribosomal and chloroplast markers used ($p = 0.01$ and $0.14$, respectively). Results from this test, however, can be affected by the disparity in levels of homoplasy between the data sets, as reported by *Dolphin et al. (2000)*. Numerous papers have discussed the limitations of the ILD to measure incongruence among data partitions (*Barker & Lutzoni, 2002*; *Dowton & Austin, 2002*; *Planet, 2006*; *Quicke, Jones & Epstein, 2007*). Imbalance between partitions is evident in this case (Table 3). Excessive type I error for the ILD test as a measure of combinability has been reported and a critical value of 0.001 has been proposed (*Cunningham, 1997*). Therefore, the ILD test should not be used as a "hard" method to decide about combinability, but as an approach to explore congruence. On the other hand, the utility of topological tests is questionable when trees from some partitions recover only a few clades with low support and no support whatever for the remaining groups (e.g., rpl16).

The tree based on all available molecular evidence preserves the geographic structure for the most part, retaining nodes with high support. Moreover, it resolves a split at the base of the subtribe, showing two clear clades containing Venezuelan and Colombian species. Both the split network (Fig. S1) and the haplotype network (Fig. S2) show similar results.
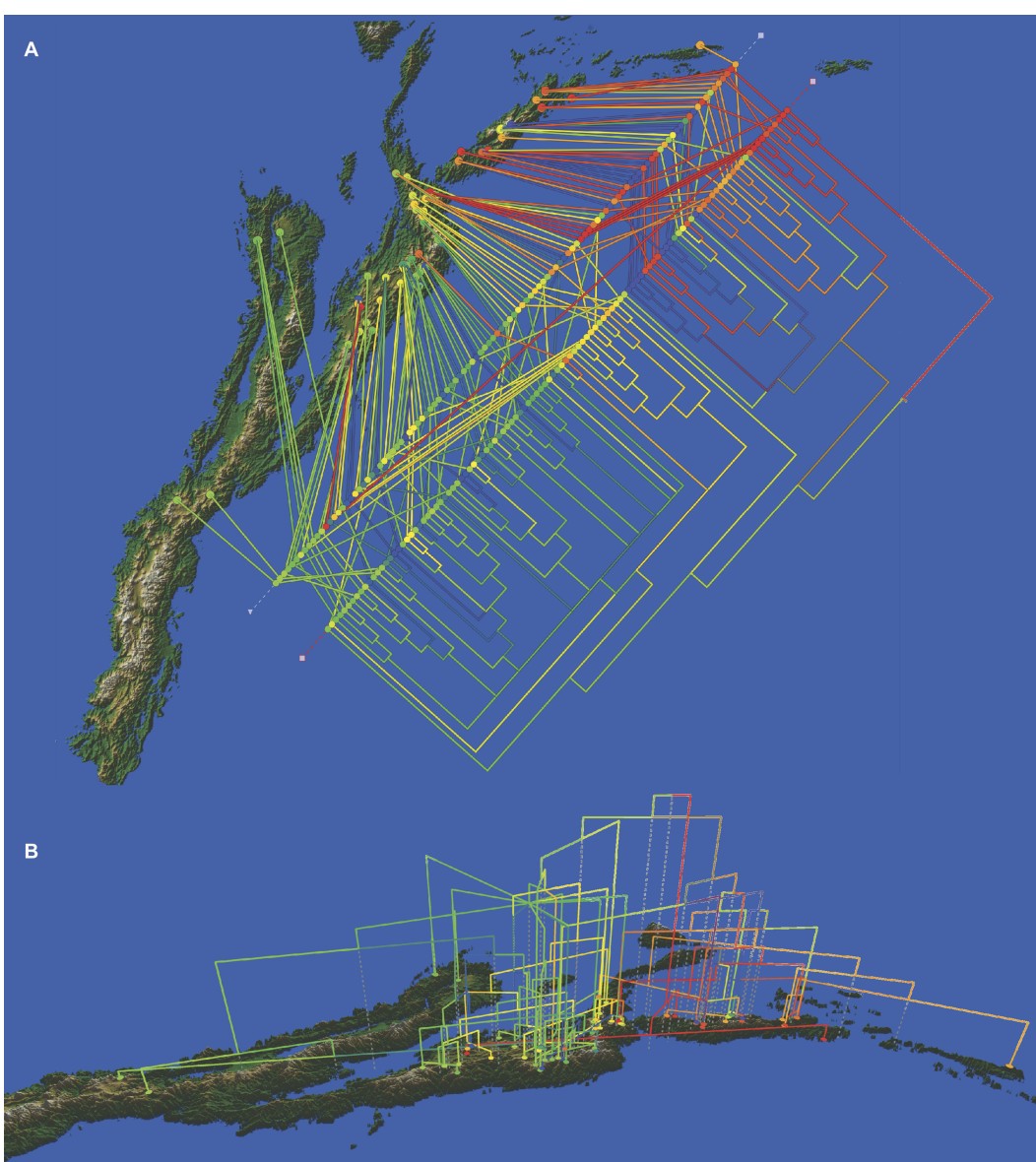

**Figure 6** **Cladogram of the Bayesian tree based on the available molecular evidence (ETS, ITS, rpl16 and AFLPs), mapped on the geographic range of the subtribe.** (A) top view of the cladogram; (B) side view of the cladogram. Additional line of colored dots in the top view shows the samples arranged by geographic proximity.

Results of this project indicate that increasing the amount of data can help to increase the phylogenetic signal, at least for shallow phylogenies of rapid radiated groups. Data from the latest high-throughput sequencing technologies should facilitate deeper exploration into the origin and evolution of such groups.

## Hybridization and evolution of frailejones

Hybrid speciation and reticulate evolution are common processes in plants, and have been reported widely for Asteraceae (*Mallet, 2007*; *Moodya & Rieseberg, 2012*; *Nolte & Tautz, 2010*). It may take several million years after the split of a species pair before the capacity to

hybridize is completely lost (*Nolte & Tautz, 2010*). Thus, species of hybrid origin in young groups may be more common than previously thought. A species of hybrid origin maintains allelic combinations that contribute to the spread and stabilization of the hybrid lineage, and it is generally recognized as a species through time (*Mallet, 2007*). Hybridization does not necessarily involve polyploidy in closely related taxa; there are other mechanisms that facilitate rapid hybrid speciation in sympatric or parapatric species, such as transgressive segregation and ecological hybrid speciation (e.g., sexual selection; *Seehausen, 2004*).

Hybridization is a frequent natural process across the Espeletiinae and it has been reported in a number of sources. *Diazgranados (2012a)* states that when two or more sympatric or parapatric species of frailejones occur, hybridization usually happens and he has documented this with a listing of the 33 published hybrids. In addition, individuals that hypothetically have introgressed with three species have been documented and numerous hybrid zones have been reported for the páramos of Mérida (*Morillo & Briceño, 2007*; *Rauscher, 2000*). It is also possible that a few putative species with unknown populations are hybrids. *Cuatrecasas (2013)* recognized the putative hybrid origin of a few taxa, and *Rauscher (2002)* confirmed eight different natural hybrid crosses involving 12 species. Hybridization has been proposed as an important mechanism in adaptive radiations (*Seehausen, 2004*) and it appears that it is important in the Expeletiinae as well.

The rapid radiation of frailejones could be explained by hypotheses of both allopatric speciation and hybrid swarm origin. The high altitudes of the tropical Andes provided multiple underutilized niches (i.e., fitness peaks) for colonization. If a colonizing species contained sufficient variation at functional loci, it could express multiple fitness peaks simultaneously. If only subsets of these peaks were effectively utilized, different populations of functional genotypes could have emerged rapidly by multiple events of ecological speciation (*Schluter, 2000*). Glaciations would have increased the probability of secondary contact between related species or divergent populations, generating a 'syngameon' scenario. Under these conditions, if hybrids had no ecological disadvantage and more functional gene combinations than the parental species, niche partitioning could have favored rapid speciation in sympatric species. Interglaciations would later segregate populations, favoring allopatric speciation.

Intermediacy or conflicting positions in the phylogenies can be explained by hybridization events or incomplete lineage sorting in recent radiations (*Knowles & Chan, 2009*; *Knowles & Carstens, 2007*). The very large amount of homoplasy found in the AFLPs data (HI = 0.841) can be explained as well by the homogenization of hybrid parental genomes at each AFLP locus (*Seehausen, 2004*).

## Generic relationships

Most of the genera proposed by *Cuatrecasas (1976)* and *Cuatrecasas (1995)* are at least partially supported in all trees. *Carramboa* is monophyletic in most trees (Figs. 3–5) except the AFLP tree (Fig. 2), where *C. rodriguezii* is placed (without support) in a different clade that includes a sympatric species, *R. marcescens*. Hybridization occurs frequently between *Carramboa* and other sympatric species (*Morillo & Briceño, 2007*; *Rauscher, 2000*) and it

is possible that the very small and patchy populations of this species are hybridizing with *R. marcescens*.

The *Carramboa* clade is subtended by *Ruilopezia josephensis* and *Espeletia weddellii*, both placements with no support (Figs. 3 and 4). The first species is likely a hybrid (J Mavárez, pers. comm., 2016, based on new genomic evidence), with scattered individuals in only one locality of the Páramo de San José, in sympatry with *Carramboa badilloi* and various species of *Ruilopezia*. *Espeletia weddellii* is a species with a somehow variable morphology that hybridizes frequently (e.g., with *E. marthae* or *E. schultzii*). Therefore its conflicting position with lack of support might be explained by the effect of introgression.

In the trees based on sequence data (Figs. 3 and 4) and all the available molecular evidence (AME; Fig. 5), *Libanothamnus* is paraphyletic because it includes *Espeletiopsis × cristalinensis* (not included in the total evidence analysis) and *Ruilopezia cardonae*. However, the former species is a hybrid between *Libanothamnus neriifolius* and *Espeletia aristeguietana* (*Diazgranados, 2012a*). With respect to the latter, *R. cardonae* is the southernmost of the *Ruilopezia* species, found in an area of less than 1 km$^2$ in the Tamá massif. This páramo is separated from the closest páramo in Venezuela by a distance of ca. ∼40 km. In between, the Táchira depression forms a deep valley of unsuitable conditions for any Espeletiinae. It is plausible that *Ruilopezia cardonae*, along with two or three *Libanothamnus* species, could have crossed this depression during a cold Pleistocene period. Since then, *R. cardonae* has been evolving isolated from its congeners, and probably hybridizing with *L. tamanus* (Cuatrec.) Cuatrec. Interestingly, both species share some morphological characteristics, such as very similar leaf tomentum and venation. Ongoing or past hybridization between these species can explain the position of *R. cardonae* within the *Libanothamnus* clade.

In the AME reconstruction (Fig. 5), three *Coespeletia* species form a clade with *Espeletia cuniculorum*, whereas in the nrDNA (Fig. 3) and sequence data (Fig. 4) trees, *E. cuniculorum* and *E. semiglobulata* (not included in Fig. 5) are sister to the *Coespeletia* clade. Both *Espeletia* species grow in sympatry with species of *Coespeletia* and are restricted to the superpáramos over 4,000 m. *Espeletia cuniculorum* is a rare species, known only from a few collections from the Páramo de los Conejos (Mérida), and it may be a hybrid species. *Espeletia semiglobulata* grows in the same massif, and its epithet refers to the semiglobular capitula, similar to the capitula of all *Coespeletia* species. Furthermore, *E. semiglobulata* shares with *C. moritziana*, *C. spicata* and *C. timotensis* characters such as large pendulous capitula, pluri- or multi-(6-7)-seriate short rays, and reduced length of pollen spines. It was originally assigned to *Espeletia* because of its thyrsoid synflorescence, but placed in its own section (sect. Badilloa Cuatrec., in *Cuatrecasas, 2013*); the molecular evidence suggests that the delimitation of *Coespeletia* must be revised.

*Coespeletia thyrsiformis* (A.C.Sm.) Cuatrec. is at the base of the clade with two *Coespeletia* species and *E. cuniculorum* in the AME tree (Fig. 5), but the sequence data reconstruction (Fig. 4) shows it in a clade with other four *Ruilopezia* species and the two *Espeletiopsis* with white-purplish flowers and thin leaves. It grows in sympatry with species of *Ruilopezia* (e.g., *R. leucactina*) in the Páramo del Batallón, ca. 100 km southwest from the páramos in Mérida, where the other species of *Coespeletia* are found. While *C. thyrsiformis* grows at ca. 3,230 m (2,500–3,510 m) of altitude, in the páramo proper, the other species of the genus grow at ca.

4,000 m., in the superpáramos. Because of its characteristic thyrsoid capitulescence, with polycephalous peduncles becoming monocephalous, *Cuatrecasas (1986)* and *Cuatrecasas (2013)* suggested that *C. thyrsiformis* could be more related to the ancestor of the genus that was presumably adapted to lower altitudes. Later migration upward with adaptation to extreme cold habitats would have produced the superpáramo species that are known today. However, an alternative explanation is that the species exhibits introgression with the sympatric *Ruilopezia* species, explaining its confusing position in the phylogeny.

A fifth species, *Coespeletia moritziana*, is also sympatric with the superpáramo species of Mérida. However, it appears at the base of a larger clade containing five different genera (Figs. 3–5). This conflicting position was also reported by *Rauscher (2000)* for ITS. He suggested two possible explanations: parallel evolution or hybridization. *Coespeletia moritziana* shares numerous morphological and phenological characteristics that place it without doubt in the genus *Coespeletia*. However, it hybridizes frequently, at least with *C. timotensis* and *E. schultzii*, and exhibits a plastic morphology. *Coespeletia moritziana* forms large well-established populations and could be a species of hybrid origin.

Sister to the *Coespeletia* clade (minus *C. moritziana*) is a small clade containing *Espeletiopsis angustifolia, E. pannosa* and *Ruilopezia floccosa*. These three species share silvery sericeous pubescence and have common names such as 'frailejón plateado' (i.e., silvery frailejón). There is no doubt about the relatedness of two of the *Espeletiopsis* species, both of which have white-purplish flowers and thin leaves. These two species show no close relationship with other species of *Espeletiopsis*, and a re-classification could be suggested for them. *Ruilopezia floccosa* grows in sympatry with *E. angustifolia*, but no large populations are currently known. It is peculiar that whereas most of the *Ruilopezia* species live in subpáramo areas at the limit of the forest, *R. floccosa* is adapted to open grass páramos. Further analyses are needed to determine the origin of this taxon.

The remaining species of *Ruilopezia* are grouped in a clade that includes *Tamania* (Figs. 3–5). Both genera have terminal synflorescences. The life form of *Ruilopezia* was defined as a monocarpic caulirosula, i.e., a rosette of imbricated leaves at the end of a straight stem, with a terminal synflorescence (*Cuatrecasas, 1933*; *Cuatrecasas, 1934*; *Cuatrecasas, 2013*). After flowering, the rosette (or the ramet) dies. *Tamania* is a tree with a monopodial trunk but sympodial (pseudodichotomous) branching, and terminal monochasial synflorescences as well. *Cuatrecasas (1986)* hypothesized a relationship between these two genera. According to him, these two genera and *Libanothamnus*, which also has terminal synflorescences, share a common ancestor. Interestingly, a clade of most of the *Ruilopezia* species + *Tamania* + *Libanothamnus* is well supported in the tree based on total evidence (Fig. 5).

Three segregated species of *Ruilopezia* (*R. leucactina*, *R. lindenii* and *R. marcescens*) form a separate well-supported clade (Figs. 3–5). *Ruilopezia lindenii* and *R. marcescens* share numerous synapomorphies: large broad herbaceous sterile phyllaries (≥20 mm long), densely glanduliferous floral tube with sparse hairs, and very long disk corollas (6–8 mm long). The three species share creamy-white (sometimes light greenish) ray flowers.

The Colombian clade contains only species of *Espeletia*, *Espeletiopsis* and *Paramiflos* (Fig. 5). *Espeletiopsis diazii*, a new species recently discovered (*Diazgranados & Sanchez,*

*2013*) is at the base of this clade. This is a unique species, particularly interesting because it shares with some Venezuelan species of *Espeletia* the oblong shape of the sheaths at the base of linear tomentose leaves, among other characteristics. However, the synflorescence is a typical monochasium that places this species under *Espeletiopsis*. Currently found in a remote páramo in Norte de Santander, it may be a descendant of one of the first species that migrated from Venezuela.

A clade containing twelve *Espeletiopsis* species suggests the monophyly of the Colombian *Espeletiopsis*. In addition to *E. diazii*, five species (*E. colombiana* (Cuatrec.) Cuatrec., *E. funckii* (Sch. Bip. ex Wedd.) Cuatrec., *E. guacharaca* (S. Díaz) Cuatrec., *E. petiolata* (Cuatrec.) Cuatrec., and *E. pleiochasia*) and one sample of *E. garciae* (Cuatrec.) Cuatrec. fall outside this clade. The positions of these species, however, are either not supported or represent relationships with sympatric species. *Rauscher (2000)* found two different ITS haplotypes for *E. garciae* and *E. pleiochasia*, suggesting either hybridization or two possible lineages for ITS. Sister to the large clade of Colombian *Espeletiopsis*, is a large clade of Colombian *Espeletia*, although support for this clade is very low. Species within this last clade are grouped mainly by their geographic location.

One sample of *Espeletia congestiflora* (Coll. number 3,651) seems to be nested with *Espeletiopsis diazii* and related to other *Espeletiopsis* taxa sister to the larger clade containing mainly *Espeletia* species (Figs. 3 and 4). *Diazgranados (2012b)* initially treated this collection as a new species to be described (tentatively named *Espeletia multicongestiflora* sp. nov. in his dissertation). However, the species has not been described yet, and can fit in the broad description of *E. congestiflora* sensu lato, which will need to be adjusted. Population of *E. congestiflora* 3,651 differs mainly from the typical *E. congestiflora* morphology in having capitulescences with 9–12 congested capitula, rather than 3–7, and alternate sterile bracts along the scape, which is typical of *Espeletiopsis* and not *Espeletia*. This putative new species has yet to be studied, and for the moment the collection was placed within the conservative definition of *E. congestiflora*.

## Evolution and geography

Our results support a recent rapid radiation of the Espeletiinae, based on a shallow phylogeny with respect to other sister species, a recent origin (less than 2–4 my BP), and a great number of species. Frequent hybridization is unmistakable, and most species may exhibit incomplete lineage sorting. Trees based on sequence data clearly support an origin of the subtribe in Venezuela, as hypothesized originally by *Cuatrecasas (1986)*.

The apparent mixing of genera in the Venezuelan clade can potentially be explained by a much longer history of introgressive hybridization. The relatively longer branches of the Venezuelan species in comparison with the Colombian species suggest older ages of those taxa. Branch length differences between putatively older Venezuelan and younger Colombian taxa (most pronounced in the combined sequence tree; Fig. 4), could perhaps reflect a progression of species migrations in time and space. Longer branches are more common among the species from the massif of the Venezuelan Mérida páramos, while species at the extremes of the geographic range of the subtribe tend to have shorter branches. Thus, the massif of the Mérida páramos could be hypothesized as a putative center of origin

for the subtribe. From there, it followed mainly southward migrations along the Andes cordilleras (*Cuatrecasas, 1986*; *Cuatrecasas, 2013*).

The Colombian clade of *Espeletiopsis* (including *Paramiflos*, Fig. 5) shows numerous relationships between sympatric or parapatric species: *E. guacharaca* and *P. glandulosus* (Cuatrec.) Cuatrec. in the páramo de la Rusia (Boyacá); *E. insignis* (Cuatrec.) Cuatrec. and *E. sclerophylla* (Cuatrec.) Cuatrec. in the páramo de Almorzadero-Chitagá (Norte de Santander); *E. caldasii* (Cuatrec.) Cuatrec. and *E. santanderensis* (A. C. Sm.) Cuatrec. in the páramo de Berlín-Almorzadero (Norte de Santander); *E. corymbosa*, *E. garciae* (col. 3715) and *E. rabanalensis* S. Díaz & Rodr.-Cabeza in the páramos of Cundinamarca-Boyacá; and *E. sanchezii* S. Díaz & Obando and *E. purpurascens* (Cuatrec.) Cuatrec in the páramo complex of Tierranegra–Tamá (Norte de Santander).

The Colombian clade of *Espeletia* species in the AME tree (Fig. 5) does not show resolution at the base; however, numerous small clades reflecting geographic distributions are recovered. Colombian species suggest two important centers of radiation for Espeletiinae: the páramos of Santander and Norte de Santander, close to Venezuela; and the páramos of Boyacá, where the Eastern Cordillera reaches its maximum width and topographic complexity. A smaller center of diversification is the complex of páramos in Cundinamarca, around the 'Sabana de Bogotá'. The AME phylogeny suggests that Colombian species of *Espeletia* from these areas are more related to each other, in disregard of their altitude or niche specialization, than with other vicariant species. As an example, *E. lopezii* appears to be related to *E. cleefii* Cuatrec. These are two parapatric species that can be found in the Sierra Nevada del Cocuy (Boyacá–Arauca). *Espeletia lopezii* prefers swampy, very wet meadows, while *E. cleefii* thrives better in scarped ridges of the superpáramo. The former has long robust naked scapes with a simple 3-cephalous cyme, while the latter has scapes with multiple pairs of sterile leaves and 15–27 capitula. There are numerous morphological differences that would classify these two species in totally different groups within *Espeletia*. Similarly, several clades containing morphologically divergent species, found in the same páramo massifs, are recognized as closely related species (i.e., clades for the páramos of Pisba, Frontino, Chingaza-Sumapáz-Tablazo, Tota, Iguaque-La Rusia, Nariño, etc.). It is likely that these clades represent the most recent colonization events that likely occurred after the Last Glacial. Additional glaciations would have enabled secondary crosses between vicariant species, with a diffusion of the geographic structure.

Geographically, the phylogeny suggests that the radiation of frailejones is an ongoing and highly dynamic process. None of the proposed two hypotheses describes entirely this radiation, but rather a combination of the two, with numerous horizontal and vertical migrations, isolations and reconnections, and a strong geographic structure. Frequent hybridization is likely prolonging the radiation momentum.

Based on morphology and his knowledge of the group, *Cuatrecasas (2013)* proposed a schematic phylogeny of the subtribe (Fig. 4), in which he highlighted two main clades. One contains *Libanothamnus*, *Ruilopezia* and *Tamania*, the group with terminal capitulescences. Our results (Figs. 3–5) support this hypothesis, with *Tamania* clearly nested with *Ruilopezia*. The other clade proposed by *Cuatrecasas (2013)* includes the rest of the genera, with *Paramiflos* closely related to *Espeletiopsis*, as well as *Coespeletia* with

*Espeletia*, and *Carramboa* as sister of the latter clade. Our phylogenetic reconstructions support the position of *Paramiflos* within *Espeletiopsis* (Figs. 3–5). However, it does not support the relationship of *Carramboa* and *Coespeletia* with the large clade of *Espeletia* (with the Colombian species of the genus). Moreover, the result of two large clades (mostly Venezuelan species and Colombian species) deeply challenges Cuatrecasas' generic classification, with two possible outcomes: a splitting approach or a lumping approach. The former would imply: (1) preserving the monophyletic *Carramboa*; (2) redefining generic limits of *Coespeletia*, likely including *Espeletia semiglobulata* and *E. cuniculorum*; (3) proposing a new generic combination for *Espeletiopsis pannosa* and *E. angustifolia*; (4) splitting *Ruilopezia* in two genera, one of them including *Tamania*, and the other one the mainly glandulous *Ruilopezia* species; (5) creating a new genus (or two) for the Venezuelan species of *Espeletia* (considered by *Cuatrecasas (2013)* as section *Weddellia* Cuatrec.); (6) splitting Colombian *Espeletiopsis* in three genera; and (7) keeping Colombian *Espeletia* species as the proper genus *Espeletia*. The lumping approach would imply creating two groups (or three if keeping the monophyletic *Carramboa*): one with the mainly Venezuelan species, in which case should be named *Libanothamnus*; and one group with the Colombian taxa, named *Espeletia*. In any case, it seems premature to address these profound changes in the classification without further analysis (e.g., genomic data using population sampling) to fully understand conflicting inter-specific relationships.

Finally, our conclusions strongly support that relationships between species cannot be established solely in the light of morphology. Behind every species there is an evolutionary hypothesis to test, and frailejones represent a fertile field for studying migration, speciation and hybridization mechanisms.

## ACKNOWLEDGEMENTS

We wish to thank Gerardo Camilo, Vicki Funk, Jason Knouft, Allison Miller and Peter Raven for enlightening discussions and constructive feedback. We are also grateful for the contributions of many undergraduate and graduate students to the molecular analyses: Christine Abboud, Emily Adamson, Anna Belia de los Santos, Rachel Fauser, Ameera Haider, Hannah King, Shaun Patel, Amith Reddy and Geetha Sridharan. Four students and colleagues deserve our most special acknowledgments for their contributions to the lab work: Talita Carvalho, Maria Pinilla, Carolina Romero and Carolina Sánchez. We thank our local collaborators in Colombia: Camilo Cadena, Santiago Madriñán and Roberto Sánchez; and in Venezuela: Gilberto Morillo and Luis 'Kike' Gamez. We especially would like to acknowledge the many friends and colleagues who contributed as field assistants: Camilo Cadena, Monica Carlsen, Andrés Diavanera, Elí Flores, Carlos Gómez, Jennifer Gruhn, Rachel Jabaily, César Alirio Leal, Llizeth Mantilla, Clara Quintero, Diego Rodríguez, Nicolás Rodríguez, Susana Rodríguez, César Sanabria, Roberto Sánchez, Jessica Sarmiento, the park rangers of Pisba, Chingaza, Tamá and Denira, and many others. Without their considerable help, this project could not have been accomplished. Additional plant samples were provided by Fernando Alzate, Rodrigo Camara, Mauricio Castilblanco, Filip Kolar, Betsy V. Rodríguez, Oscar Salazar, Petr Sklenář and Simón Uribe-Convers. Special thanks

to the institutions and herbaria that opened to the authors their collections, especially: ANDES, CAS, COL, CUVC, F, FMB, HECASA, HUA, K, MER, MO, NY and US. We thank Carlos Parra, from COL, and Santiago Madriñán, from ANDES, for providing working and storage facilities at their respective herbaria in Colombia. Special thanks to Monica Carlsen for her comments and advice on the phylogenetic analyses. We also thank the Missouri Botanical Garden friends and colleagues for their comments and support.

### Funding

Financial support provided by Saint Louis University, the National Science Foundation (as a Doctoral Dissertation Improvement Grant, DEB 1011624), the Committee for Research and Exploration of the National Geographic Society (Grant No. 8613-09), the Missouri Botanical Garden, Smithsonian Institution (SI) graduate fellowship, SI Cuatrecasas Fellowship, SI Latino Initiative, Society for Systematic Biologists, American Society of Plant Taxonomist, Botanical Society of America, Idea-Wild Foundation, the Neotropical Grassland Conservancy and the Fondo Colombia Biodiversa of the Fundación Alejandro Ángel Escobar. The funders had no role in study design, data collection and analysis, decision to publish, or preparation of the manuscript.

### Grant Disclosures

The following grant information was disclosed by the authors:
Saint Louis University.
National Science Foundation: DEB 1011624.
Committee for Research and Exploration of the National Geographic Society: 8613-09.
Missouri Botanical Garden.
Smithsonian Institution (SI) graduate fellowship.
SI Cuatrecasas Fellowship.
SI Latino Initiative.
Society for Systematic Biologists.
American Society of Plant Taxonomist.
Botanical Society of America.
Idea-Wild Foundation.
Neotropical Grassland Conservancy.
the Fondo Colombia Biodiversa of the Fundación Alejandro Ángel Escobar.

### Competing Interests

The authors declare there are no competing interests.

### Author Contributions

- Mauricio Diazgranados conceived and designed the experiments, performed the experiments, analyzed the data, contributed reagents/materials/analysis tools, wrote the paper, prepared figures and/or tables.

- Janet C. Barber contributed reagents/materials/analysis tools, reviewed drafts of the paper, advised the research process.

## Field Study Permissions

The following information was supplied relating to field study approvals (i.e., approving body and any reference numbers):

Collections and DNA extractions were made under permits No. 2698 of 09/23/2009 and No. 2 of 02/03/2010 (Ministerio de Ambiente, Colombia), and IE-126 (Venezuela, authorized by Petr Sklenář).

## DNA Deposition

The following information was supplied regarding the deposition of DNA sequences:

Sequences available in GenBank with accession numbers: ETS: KY231675–KY231818, ITS: KY231383–KY231532 and rpl16: KY231533–KY231674. GenBank accession numbers and voucher information per sample are provided in Table S1.

## Data Availability

The raw data has been supplied as a Supplementary File.

## Supplemental Information

Supplemental information for this article can be found online at http://dx.doi.org/10.7717/peerj.2968#supplemental-information.

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
