# Peer review of "Geography shapes the phylogeny of frailejones (Espeletiinae Cuatrec., Asteraceae): a remarkable example of recent rapid radiation in sky islands"

_PeerJ, doi:10.7717/peerj.2968_

## Round 0.1 · original submission · Minor Revisions

Please make corrections where you agree with the reviewers' comments, or explain your decision where you do not.

Reviewer 1 ·

Basic reporting

No comments

Experimental design

No comments

Validity of the findings

I would like to see more details in “generic relationships” section.
1. Pay more attention to the tree on fig. 4 rather then to the one on fig. 5. According my experience combined tree based on ITS, ETS and rpl16 is more appropriate for discussion on generic limits rather then AME which is based also on ALFP (fig. 5), because the latter marker is used mainly for complexes of closely related species and infraspecific relationships.
2. It would be appropriate to discuss the possibility of changing generic limits according the tree on fig. 4. — to move some species from one genus to another and to make corresponding nomenclatural combinations. If this is premature, why?
3. It is recommended to discuss specially the situation with the largest genus Espeletia which is paraphyletic. E.g., does Espeletia congestiflora (nested on fig. 4 within the main part of Espeteliopsis) have some unique morphological characters or not? Are there grounds to move it to Espeteliopsis? Similar situation with Espeletia weddellii, etc.
Also it would be reasonable to pay more attention on incongruence between trees based of different markers, e.g. specially discuss most important examples.

Additional comments

Your paper is very interesting and definitely merits publication. However, you data could be used not only for conclusions about 2 centers of diversity and that “the radiation of frailejones is an ongoing and highly dynamic process”. You can make suggestions (or at least speculations) on generic limits and status of some species (see comments in “Validity of the findings”). Some minor comments see also in attached file.

Annotated reviews are not available for download in order to protect the identity of reviewers who chose to remain anonymous.

·

Basic reporting

No comment

Experimental design

No comment

Validity of the findings

No comment

Additional comments

The research fit well within Scope of journal. Methods of DNA extraction and phylogenetic reconstruction are carefully described, there are enough information for replication of some author's modifications. The results are based on study both numerous specimens collected by authors and specimens preserved in different Herbaria. Constructed trees are discussed in detailes. There are no special conclusion section but last section "Evolution and geography" can serve as conclusion.
The paper represents an example of thorough research. Introduction is large and includes both brief descriptions of paramo ecosystem and taxonomy of frailejones. The research is based on study of numerous specimens both collected by authors and herbarium sheets from numerous herbaria. Methods are described in detailes. Apart of standart technics authors use some modifications that are well described. Explanation of rapid radiation of frailejones is appropriate. All statements are well supported. Some abiquity in constructed trees are discussed. English is clear and as far as I can judge professional. The paper is easy to read for not native speakers. It was pleasure for me to read this paper in spite of my own interests are far away from paramo and frailejones.

---

## Round 0.2 · accepted · Accept

The reviewers' concerns have been addressed.